# Aerobic capacity and muscular strength in 179 Norwegian men and women aged 20–59 years with a substance use disorder: A cross-sectional study

Henrik Loe[1,2*], Grete Flemmen[1,3], Ulrik Wisløff[4], Helle W. Andersson[1], Mats P. Mosti[1]

1 Department of Research and Development, Clinic of Substance Use and Addiction Medicine, St. Olav's University Hospital, Trondheim, Norway, 2 Department of Public Health and Nursing, Faculty of Medicine and Health Sciences, Norwegian University of Science and Technology, Trondheim, Norway, 3 Department of Mental Health, Faculty of Medicine and Health Sciences, Norwegian University of Science and Technology, Trondheim, Norway, 4 Department of Circulation and Medical Imaging, Faculty of Medicine and Health Sciences, Norwegian University of Science and Technology, Trondheim, Norway

* henrik.loe@stolav.no

## Abstract

Research has shown a robust inverse association between maximal aerobic capacity and muscular strength, and lifestyle related diseases and mortality. People with a substance use disorder have a higher prevalence of lifestyle-related diseases and a shorter life expectancy than the general population. There is a paucity in the literature of normative aerobic capacity and muscular strength variables in people with a substance use disorder. The main purpose of this study was to contribute to establish reference values on these key variables for this patient group, and to compare these data to normative values consisting of healthy age matched controls. A total of 179 men and women with diagnosed substance use disorder according to International Classification of Diseases-10 performed a maximal cardiopulmonary exercise test on a treadmill recording maximal oxygen uptake and maximal oxygen pulse, along with a muscular strength test in a hack squat machine assessing maximal leg strength. Patients with substance use disorder generally demonstrated lower- maximal oxygen uptake (∼ 15%), maximal oxygen pulse (∼ 10%) and muscular strength (∼ 30%) across all age groups compared to comprehensive Norwegian and American reference materials of healthy age-matched populations. This could contribute to explain the higher prevalence of lifestyle related diseases and reduced longevity in this patient group.

## Introduction

Maximal oxygen uptake ($VO_{2max}$), the highest rate at which oxygen can be transported and utilized by the cardiovascular system and peripheral tissues [1–3], has consistently demonstrated a robust inverse association with prevalence of

**Data availability statement:** All relevant data are within the paper and its Supporting Information files.

**Funding:** This work is supported by Ekstrastiftelsen Helse og Rehabilitering (Stiftelsen DAM) grant number 2019/F0235306 and The Joint Research Committee between St.Olavs hospital and the Faculty of Medicine and Health Sciences, Norwegian University of Science and Technology grant number 2023/36515. The funders had no role in study design, data collection and analysis, decision to publish, or preparation of the manuscript.

**Competing interests:** The authors have declared that no competing interests exist.

non-communicable diseases (i.e., cardiovascular disease [4–9], metabolic syndrome [10–13] certain cancers [14–17], type II diabetes [18,19], hypertension [20,21] and all-cause mortality [4,22–27]. Maximal Oxygen pulse (max $O_{2pulse}$), a noninvasive surrogate measure of left ventricular stroke volume that denotes oxygen pumped per heartbeat [28], also possess significant prognostic value for cardiovascular disease as well as being a potent mortality predictor [29]. More recently, muscular strength has emerged as an important forecaster of somatic health, showing inverse association with cardiovascular disease [30,31], type II diabetes [32], cancer [33], as well as serving as an independent predictor of all-cause mortality [31,34–37].

Individuals with substance use disorder (SUD) show a higher prevalence of lifestyle-related diseases [38–41], and their life expectancy is estimated to be 15–30 years shorter than that of the general population. This reduced longevity is largely attributed to poor mental and somatic health [39,41,42]. Findings indicate that a significant proportion of individuals with a SUD lead a sedentary lifestyle [39], with insufficient levels of physical activity [43]. This lack of activity has a direct negative impact on both aerobic capacity and muscular strength.

There is a notable paucity in the literature of objectively measured reference materials on these key variables in patients with a SUD. The main purpose of this study was to contribute to establish reference values for directly measured aerobic capacity- and muscular strength variables in patients with SUD, and to compare these data to normative values in the HUNT3 cohort consisting of apparently healthy age matched controls.

## Materials and methods

### Study design

This clinical study utilized cross-sectional baseline data from an ongoing prospective cohort study conducted at the Clinic of Substance use and Addiction Medicine, St Olav's University Hospital, Trondheim, Norway. The study was approved by the Regional Committee for Medical and Health Research Ethics (2019/501/REK midt), registered in Clinical Trials (NCT04141202), and carried out in compliance with the Declaration of Helsinki (WMA Declaration of Helsinki 2013). Participants were insured by the Norwegian patient compensation claim.

### Settings and participants

This study included, 179 inpatients (26% women) aged 20−59 years, diagnosed with SUD in accordance with the International Classification of Diseases, 10th Revision (ICD-10, F10-19) [44]. The primary SUD diagnosis is detailed in Table 1. Participants were recruited successively upon admittance to the clinic of Substance Use and Addiction Medicine, St. Olavs University Hospital, Trondheim, Norway. A detailed study description (oral and written) was provided to eligible volunteers, and informed consent was secured. Inclusion criteria: being able to perform exhausting endurance- and strength training and all-out testing, not been abstinent from drugs last 6 months, being in residential treatment at our clinic, free from known cardiovascular- or

**Table 1. Primary diagnosis.**

|  | Men | Women |
|---|---|---|
| F10 | 39.4% | 38.3% |
| F11 | 12.9% | 10.0% |
| F12 | 18.1% | 15.0% |
| F13 | 4.5% | 8.3% |
| F14 | 4.5% | 5% |
| F15 | 20.0% | 23.3% |
| F19 | 0.6% | 0.0% |
| Current smoker | 52.3% | 56.7% |
| Polydrug use | 64.5% | 65.0% |

Data is presented as percent of inpatient population. Mental- and behavioral disorders caused by- F10: alcohol, F11: opiates, F12: cannabinoids, F13: sedatives and hypnotics, F14: cocaine, F15: other stimulants includingcaffeine, F19: multiple substances or other psychoactive substances (ICD-10) Polydrug use: self-reported use of multiple substances.

pulmonary disease and cancer. Patients that fulfilled the inclusion criteria, and agreed to participate, were tested and measured onsite in our clinic. All participants successfully completed both the $VO_{2max}$ and maximal muscular strength test in compliance with preset max criteria (specified under the heading: Physical testing and measurements). No adverse events occurred. Of 207 eligible candidates, 28 declined to participate, resulting in an ∼87% inclusion rate. Recruitment commenced January 1st, 2020, with conclusion December 22st, 2023, paused from mid-March 2020 through primo October 2020 due to the Covid-19 pandemic.

## Physical testing and measurements

Resting heart rate ($HR_{rest}$) was measured, using a SunTech Tango M2 (SunTech Medical, Morrisville, NC, USA), sitting in an upright position subsequent a 5-minute resting period. Bodyweight was measured with an M-420 Marsden (Marsden Weighing Machine Group Ltd, Rotterham, England).

## Cardiopulmonary exercise testing (CPET)

Detailed information on the CPET procedure was provided before the test. We used a PPS 55 Med treadmill (Woodway USA, Waukesha, WI, USA) for the all-out physical performance test. HR data and respiratory kinetics were continuously recorded using Polar H7 (Polar Elektro Oy, Kempele, Finland) and a Vyntus CPX fitted with an ambient air mixing chamber (Jaeger, Hockenberg, Germany), respectively. A 10- minute treadmill warmup and familiarization phase preceded the all-out test. Participants were instructed not to grab the handrails, if not necessary, as it would reduce the oxygen cost at a fixed workload. Warmup workload was individually adjusted to induce a moderate increase in HR. Initial treadmill velocity during test was determined based on HR, subjectively perceived exertion and workload during warmup. Treadmill velocity was increased by one km·hour$^{-1}$ per minute until the participants reached volitional exhaustion, and the test was terminated. Slope inclination remained fixed at five % throughout the test. Seven participants, unable to run, utilized a test protocol where slope inclination was increased by two % per minute and the velocity remained fixed. $VO_{2max}$/$VO_{2peak}$ was logged as the average of the three highest sequential $VO_2$ measurements. Attaining $VO_{2max}$ presupposed complying with the following criteria. (I) Observation of a stable $VO_2$ plateau, even with an increased workload [45], (II) reaching volitional exhaustion and (III) registering a respiratory exchange ratio (R) of 1.05 or higher coinciding with the time of measuring $VO_{2max}$ [46]. The CPET equipment (Vyntus CPX, Jaeger, Hoechberg, Germany) was validated against an iron lung (Metabolic Simulator with mass flow controller, Vacumed, Ventura, CA, USA). An inconsistency was observed in the highest

minute ventilations ($V_E > 180$ Lmin$^{-1}$), presenting with a 5.5% error (being in excess of the predefined acceptable error of 5%). No relevant deviations are expected in this minute ventilation range among SUD patients based on previous observations [47]. Two participants displayed $V_E > 180$ L/min (184 L/min and 213 L/min), which were unlikely to influence the results. All other measurements were inside acceptable range. The Vyntus CPX was calibrated prior to testing each day. A multistep process was performed, (1) gas calibration using a gas mix (15.99% $O_2$ and 5.00% $CO_2$ mixed with $N_2$) and ambient air, (2) calibrating the Digital Volume Transducer (DVT). A default volume calibration was performed between each test. The PPS 55 Med treadmill was calibrated (velocity (km·hour$^{-1}$ and slope angle (%)) prior to study start and controlled regularly.

### HUNT 3 reference material

**Test specifics.** The HUNT 3 study utilized a very similar protocol, calibration procedures and direct $VO_2$ measurement as in the present study and a CPET was accepted as $VO_{2max}$ if test was terminated upon volitional exhaustion, $R \geq 1.05$ and the registering of a stable $VO_2$ plateau despite increased workload. Cortex Metamax II (Cortex, Leipzig, Germany) with a built-in mixing chamber was used. Characteristics of the HUNT 3 reference material are presented in Table 2.

### Risks in using CPET

A maximal CPET is regarded as safe practice. Nevertheless, adverse events and severe complications occur [49,50]. Incidents of sudden death is reported in the range of 0–5 for every 100,000 tests, in mixed populations of apparently healthy and patients [51,52]. Risk of adverse occurrences, including death and potentially life-threatening events (e.g., ventricular fibrillation and myocardial infarction) increased to 6–10/100,000 when including high-risk patients [53,54]. Even so, CPET it is considered safe [55]. American College of Sports Medicine (ACSM) and the American Heart Association (AHA) advocate not overstating potential ill health effects related to vigorous exercise, since health benefits significantly overshadows the risks [56]. However, being observant to symptoms indicative of test termination is paramount (e.g., acute chest pain, great dyspnea, mental confusion, loss of coordination and sudden paleness) [49]. Trained staff conducted CPET.

### Oxygen pulse ($O_{2pulse}$)

Maximal $O_{2pulse}$ was calculated from the three consecutive highest $VO_2$ measurements (i.e., $VO_{2max}$) divided by the corresponding HR measurements.

**Table 2. Characteristics of the HUNT 3 reference material [48].**

|  | $VO_{2max}$ (mL·kg$^{-1}$·min$^{-1}$) | Body weight (kg) | R |
|---|---|---|---|
| **Men:** |  |  |  |
| 20-29 years (n = 199) | 54.4 ± 8.4 | 80.1 ± 10.6 | 1.15 ± 0.05 |
| 30-39 years (n = 324) | 49.1 ± 7.5 | 86.8 ± 12.1 | 1.15 ± 0.05 |
| 40-49 years (n = 526) | 47.2 ± 7.7 | 86.4 ± 11.5 | 1.15 ± 0.05 |
| 50-59 years (n = 466) | 42.6 ± 7.4 | 86.4 ± 10.3 | 1.14 ± 0.05 |
| **Women:** |  |  |  |
| 20-29 years (n = 215) | 43.0 ± 7.7 | 65.5 ± 10.4 | 1.15 ± 0.05 |
| 30-39 years (n = 359) | 40.0 ± 6.8 | 69.7 ± 11.5 | 1.15 ± 0.05 |
| 40-49 years (n = 493) | 38.4 ± 6.9 | 69.9 ± 11.2 | 1.15 ± 0.05 |

Data are presented as arithmetic mean ± SD. $VO_{2max}$: maximal oxygen uptake, R: respiratory exchange ratio.

### Maximal muscular strength testing (1RM)

Participants performed a maximal strength test using muscles in the lower extremities. Major muscles involved being m. quadriceps, m. hamstrings and m. gluteus maximus. One-repetition maximum (1RM), the highest achievable torque or force produced in a maximal voluntary concentric muscle contraction [57], was measured in a hack squat machine (Impulse Fitness IT7006, Impulse (Qingdao) Health Tech, Shandong, China). The hack squat movement cycle: (1) starting in an upright position – (2) eccentric muscle contraction braking downward movement and briefly stopping at a 90° knee joint angle – (3) returning to initial upright position by concentric muscle contraction. Warm-up consisted of two sets of five repetition at 50%, and one set of 70% of expected 1RM based on observation. The initial test-load was set somewhat lower than anticipated 1RM, with 5–10 kg load increments per subsequent attempt, expecting real 1RM achieved in 6–9 attempts [58]. The highest completed load lifted was registered as 1RM.

### Allometric scaling

Allometric scaling was used to more accurately compare $VO_{2max/peak}$ and 1RM values between individuals with different body size, since neither the oxygen cost of running at a fixed workload nor muscular strength increases in linear proportion to bodyweight [59–61]. It has been shown that inter-individual comparison of $VO_{2max}$ is more appropriate when bodyweight is raised to the power of 0.75 (i.e. $mL \cdot kg^{-0.75} \cdot min^{-1}$) [59,60]. Likewise, muscular strength should be conveyed as bodyweight raised to the power of 0.67 (i.e., $kg \cdot m_b^{-0.67}$) [61]. Failing to utilize allometric scaling in expressing relative- strength and endurance will result in a consistent over- and under estimation of performance in lighter- and heavier persons, respectively [61].

### Statistical analysis

The assumption of normally distributed data was supported by QQ-plots, variance homogeneity in key variables was supported by Levene's test. Thus, parametric statistics were applied for the analysis. Data are presented as arithmetic mean±standard deviation or 95% confidence interval, Eta-squared is reported for effect sizes. To maintain the probability of a Type I error at < 5% (0.05) [62] an analysis of variance (One-way Anova), with a Bonferroni post-hoc test was used to correct for multiple comparisons. A two-sided Independent-Samples T-test was used to identify differences between men and women, with $p < 0.05$ viewed as statistically significant. Regression analysis was deployed to improve methodological rigor by displaying the impact of anthropometrical variables on the dependent variable ($VO_{2max}$). Pearson correlation coefficient was used to measure strength between variables. SPSS 29.0.1 (Statistical Package for Social Sciences, Chicago, IL, USA) and GraphPad Prism 10.1.2. (GraphPad Software, San Diego, CA, USA) were used for data analysis and graphics.

## Results

### Sex differences in $VO_{2max}$, maximal $O_2$pulse, $HR_{rest}$ and 1RM

Women had a 15% ($p < 0.001$), 32.1% ($p < 0.001$) and 19.5% ($p < 0.001$) lower relative, absolute and scaled $VO_{2max}$ than men, respectively ($36.4 \pm 7.1$ $mL \cdot kg^{-1} \cdot min^{-1}$ vs. $42.8 \pm 7.1$ $mL \cdot kg^{-1} \cdot min^{-1}$, $2.43 \pm 0.39$ $L \cdot min^{-1}$ vs. $3.58 \pm 0.52$ $L \cdot min^{-1}$ and $103.8 \pm 17.3$ $mL \cdot kg^{-0.75} \cdot min^{-1}$ vs. $129.0 \pm 18.0$ $mL \cdot kg^{-0.75} \cdot min^{-1}$). Maximal $O_{2pulse}$ was 31.1% ($p < 0.001$) lower among women than men ($13.3 \pm 2.1$ $mL \cdot beat^{-1}$ vs. $19.3 \pm 2.9$ $mL \cdot beat^{-1}$). There were no significant sex differences in $HR_{max}$ or $HR_{rest}$ (women: $190 \pm 11$/ $85 \pm 12$; men: $192 \pm 13$/ $82 \pm 13$). Women obtained 38.2% ($p < 0.001$) and 28.2% ($p < 0.001$) lower 1RM and scaled 1RM strength, respectively, compared to men ($70.8 \pm 26.4$ kg vs. $114.5 \pm 36.3$ kg and $4.25 \pm 1.76$ $kg \cdot m_b^{-0.67}$ vs. $5.92 \pm 1.89$ $kg \cdot m_b^{-0.67}$) (Table 3).

### Age group differences in $VO_{2max}$, maximal $O_2$pulse and 1RM among men

Among men the highest relative- and scaled $VO_{2max}$ ($45.9 \pm 5.2$ $mL \cdot kg^{-1} \cdot min^{-1}$ and $136.7 \pm 13.1$ $mL \cdot kg^{-0.67} \cdot min^{-1}$) was observed in the youngest age group (20–29 years), with a ~10% ($p < 0.001$) lower average $VO_{2max}$ per subsequent decade

**Table 3. Descriptive and physiological data for SUD inpatients.**

| | All | Men | Women |
|---|---|---|---|
| | (n = 179) | (n = 133) | (n = 46) |
| Age (years) | 32.1 ± 9.1 | 32.8 ± 9.0 | 30.0 ± 9.1 |
| **Anthropometrical data:** | | | |
| Body mass (kg) | 81.3 ± 18.0 | 85.5 ± 16.7 | 69.0 ± 16.1 |
| Height (m) | 1.77 ± 0.10 | 1.81 ± 0.07 | 1.66 ± 0.08 |
| BMI ($kg \cdot m^{-2}$) | 25.9 ± 5.0 | 26.2 ± 4.9 | 25.0 ± 5.3 |
| $VO_{2max}$ ($mL \cdot kg^{-1} \cdot min^{-1}$) | 41.1 ± 7.6 | 42.8 ± 7.1 | 36.4 ± 7.1 |
| $VO_{2max}$ ($L \cdot min^{-1}$) | 3.28 ± 0.70 | 3.58 ± 0.52 | 2.43 ± 0.39 |
| $VO_{2max}$ ($mL \cdot kg^{-0.75} \cdot min^{-1}$) | 122.5 ± 21.0 | 129.0 ± 18.0 | 103.8 ± 17.3 |
| $O_2$pulse ($mL \cdot beat^{-1}$) | 17.8 ± 3.8 | 19.3 ± 2.9 | 13.3 ± 2.1 |
| $HR_{max}$ ($beats \cdot min^{-1}$) | 191 ± 12 | 192 ± 13 | 190 ± 11 |
| $HR_{rest}$ ($beats \cdot min^{-1}$) | 83 ± 13 | 82 ± 13 | 85 ± 12 |
| R ($VCO_2 \cdot VO_2^{-1}$) | 1.21 ± 0.07 | 1.21 ± 0.07 | 1.20 ± 0.09 |
| | (n = 167) | (n = 123) | (n = 44) |
| 1RM (kg) | 103.0 ± 39.0 | 114.5 ± 36.3 | 70.8 ± 26.4 |
| 1RM ($kg \cdot mb^{-0.67}$) | 5.48 ± 2.00 | 5.92 ± 1.89 | 4.25 ± 1.76 |

Data are presented as arithmetic mean ± SD. BMI: body mass index, $VO_{2max}$: maximal oxygen uptake, $O_2$pulse: oxygen uptake per heartbeat, HR: heart rate, R: respiratory exchange ratio, 1RM: 1 repetition maximum concentrically.

throughout the age groups. Between the two most senior age groups (40–49 years and 50–59 years) a steeper deterioration gradient of ∼ 18% (p = 0.02) was observed. The absolute $VO_{2max}$ was 3.65 ± 0.50 $L \cdot min^{-1}$ with no significant difference between subsequent age groups. The maximal $O_2$pulse was 19.9 ± 3.9 $mL \cdot beat^{-1}$, with no significant differences between any age groups. The highest $HR_{max}$ (197 ± 9 $beats \cdot min^{-1}$) was found in the youngest age group (20–29 years), whereupon an average ∼ 5% (p < 0.001) lower $HR_{max}$ was observed between each succeeding age group. We observed no differences in $HR_{rest}$ between consecutive age groups. On average body mass was ∼ 8% (p < 0.001) higher with each following decade, with the highest body mass (100.4 ± 17.6 kg) found in the most senior age group (50–59 years). 1RM was 120.5 ± 35.1 kg in the youngest age group (20–29 years), with no significant difference between the following age groups. The highest scaled 1RM (6.36 ± 1.80 $kg \cdot m_b^{-0.67}$) was in the youngest age group (20–29 years), with an average 13% (p = 0.002) lower scaled 1RM observed between subsequent decades (Table 4). Overall effect sizes Eta-squared for relative and scaled $VO_{2max}$, respectively ($\eta^2$ = 0.329; 0.327, p < 0.05), and scaled 1RM ($\eta^2$ = 0.108, p < 0.05).

### Age group differences in $VO_{2max}$, maximal $O_2$pulse and 1RM among women

Among women the highest relative- and scaled $VO_{2max}$ (38.4 ± 6.1 $mL \cdot kg^{-1} \cdot min^{-1}$ and 108.7 ± 14.7 $mL \cdot kg^{-0.67} \cdot min^{-1}$) was observed in the youngest age group (20–29 years), with an average ∼ 11% (p = 0.006) and ∼10% (p = 0.005) lower $VO_{2max}$ per subsequent decade, respectively. The absolute $VO_{2max}$ was 2.51 ± 0.40 $L \cdot min^{-1}$ with no significant difference between subsequent age groups. The maximal $O_2$pulse was 13.4 ± 2.1 $mL \cdot beat^{-1}$, with no significant differences between age groups. The highest $HR_{max}$ (194 ± 10 $beats \cdot min^{-1}$) was found in the youngest age group (20–29 years), whereupon an average ∼ 4% (p < 0.001) lower $HR_{max}$ was observed between each succeeding age group. We observed no differences in $HR_{rest}$ between consecutive age groups. The highest body mass (76.2 ± 15.8 kg) was found in the oldest age group (40–49 years), with no significant difference between age groups. 1RM and scaled 1RM was 74.8 ± 25.4 kg and 4.55 ± 1.70 $kg \cdot mb^{-0.67}$ in the youngest age group (20–29 years), with no significant difference between the two youngest age groups (20–29 years and 30–39 years), whereupon ∼ 42% (p = 0.016) and ∼ 48% (p = 0.011) lower values were observed

**Table 4. Physiological and anthropometrical data stratified by age groups in male SUD inpatients.**

| | 20-29 years | 30-39 years | 40-49 years | 50 −59 years |
|---|---|---|---|---|
| | (n = 58) | (n = 50) | (n = 14) | (n = 11) |
| $VO_{2max}$ (mL·kg$^{-1}$·min$^{-1}$) | 45.9 [44.5-47.3] | 42.8 [40.7-44.5] | 38.5 [35.7-42.1] | 31.5 [28.2-34.7] |
| $VO_{2max}$ (L·min$^{-1}$) | 3.65 [3.5-3.8] | 3.59 [3.4-3.7] | 3.58 [3.2-4.1] | 3.13 [2.7-3.5] |
| $VO_{2max}$ (mL·kg$^{-0.75}$·min$^{-1}$) | 136.7 [133.3-140.1] | 129.2 [123.9-133.3] | 119.3 [111.2-130.7] | 99.2 [89.5-109.0] |
| $O_2$pulse (mL·beat$^{-1}$) | 19.3 [18.5-20.1] | 19.4 [18.7-20.0] | 19.9 [17.8-22.7] | 18.5 [16.5-20.4] |
| $HR_{max}$ (beats·min$^{-1}$) | 197 [195-200] | 192 [188-195] | 184 [180-189] | 171 [163-179] |
| $HR_{rest}$ (beats·min$^{-1}$) | 84 [80 −88] | 81 [77−84] | 84 [78−89] | 79 [73−84] |
| R ($VCO_2$·$VO_2^{-1}$) | 1.22 [1.20-1.24] | 1.20 [1.18-1.22] | 1.19 [1.15-1.25] | 1.18 [1.13-1.24] |
| Body mass (kg) | 80.6 [76.8-84.4] | 85.8 [81.0-90.3] | 93.3 [84.3-104.4] | 100.4 [88.6-112.2] |
| BMI (kg·m$^{-2}$) | 25.1 [23.8-26.2] | 26.0 [24.6-27.3] | 28.2 [26.4-30.3] | 30.5 [27.3-33.7] |
| | (n = 55) | (n = 46) | (n = 12) | (n = 10) |
| 1RM (kg) | 120.5 [111.0-130.0] | 113.8 [102.3-124.5] | 110.4 [91.1-132.5] | 89.0 [65.6-112.4] |
| 1RM (kg·mb$^{-0,67}$) | 6.36 [5.88-6.85] | 5.90 [5.32-6.45] | 5.48 [4.39-6.64] | 4.06 [3.09-5.02] |

Data are presented as arithmetic mean and 95% confidence interval. VO2max: maximal oxygen uptake, O2pulse: oxygen uptake per heartbeat, HR: heart rate, R: respiratory exchange ratio, BMI: body mass index, 1RM: 1 repetition maximum concentrically.

between the two oldest age groups, respectively (Table 5). Overall effect sizes (Eta-squared) for relative and scaled $VO_{2max}$, respectively ($\eta^2 = 0.111$; 0.119, $p < 0.05$), and 1RM and scaled 1RM, respectively ($\eta^2 = 0.034$; 0.044, $p < 0.05$).

## Regression analysis of $VO_{2max}$

Age and BMI are dominant independent predictors for $VO_{2max}$ in our data, whereas smoking status has miniscule predicting power (Table 6).

## Association between $VO_{2max}$, maximal $O_{2pulse}$, 1RM and age in men and women

For men there was a weak (r = − 0.26, p = 0.003) and strong (r = − 0.58, p < 0.001) negative association between $VO_{2max}$ (absolute and relative, respectively) and age (Fig 1A). The corresponding negative correlations for women were moderate (r = − 0.32, p = 0.03) and (r = − 0.49, p < 0.001) (Fig 1B). No correlation was observed between maximal $O_{2pulse}$ and age in men (r = − 0.04, non-significant) (Fig 2A) or women (r = − 0.15, non-significant) (Fig 2B). There was a weak (r = − 0.23, p = 0.01) and moderate (r = − 0.36, p = 0.02) negative correlation between 1RM and age in men (Fig 3A) and women, respectively (Fig 3B).

## Discussion

The present study constitutes the most comprehensive reference material of objectively measured aerobic capacity- and muscular strength variables in a cohort of SUD patients. Key findings underscore this group's substantially lower cardiorespiratory fitness ($VO_{2max}$) and muscular strength (1RM) compared to age-matched healthy controls, consequently, placing SUD patients at increased risk of lifestyle related disease and all-cause mortality [24,26,33,35].

Previous findings generally demonstrate a higher $VO_{2max}$ in Norwegian [48,63] and former Scandinavian healthy populations [45,64,65] than in comparable international reference materials [66–73]. Nomadic Lapps [74] tending reindeer, and contemporary hunter-gatherer societies [75] present with $VO_{2max}$ values coinciding well with the Scandinavian data, suggesting that a higher $VO_{2max}$ may be attributed to a more physically active living, as well as genetic predispositions [76]. To maintain the integrity of the analysis, methodology should be identical when comparing results from different studies (i.e., the use of treadmill as test modality and CPET utilizing a mixing chamber). Failing

**Table 5. Physiological and anthropometrical data stratified by age groups in female SUD inpatients.**

| | 20-29 years | 30-39 years | 40-49 years |
|---|---|---|---|
| | (n = 27) | (n = 11) | (n = 8) |
| $VO_{2max}$ (mL·kg$^{-1}$·min$^{-1}$) | 38.4 [36.0-40.8] | 36.5 [31.0-41.9] | 29.7 [25.5-34.0] |
| $VO_{2max}$ (L·min$^{-1}$) | 2.51 [2.35-2.67] | 2.39 [2.18-2.60] | 2.24 [1.87-2.61] |
| $VO_{2max}$ (mL·kg$^{-0.75}$·min$^{-1}$) | 108.7 [102.9-114.5] | 103.5 [90.3-116.6] | 87.5 [76.1-98.8] |
| $O_2$pulse (mL·beat$^{-1}$) | 13.4 [12.6-14.3] | 13.2 [12.2-14.3] | 13.1 [10.8-15.4] |
| $HR_{max}$ (beats·min$^{-1}$) | 194 [190-198] | 189 [184-194] | 179 [170-187] |
| $HR_{rest}$ (beats·min$^{-1}$) | 87 [82 –92] | 82 [75 –88] | 79 [68-90] |
| R ($VCO_2$·$VO_2^{-1}$) | 1.19 [1.16-1.22] | 1.23 [1.18-1.27] | 1.21 [1.11-1.31] |
| Body mass (kg) | 67.1 [60.8-73.3] | 68.3 [56.8-79.9] | 76.2 [62.9-89.4] |
| BMI (kg·m$^{-2}$) | 24.3 [22.2-26.4] | 25.5 [21.9-29.2] | 27.0 [22.9-31.1] |
| | (n = 26) | (n = 10) | (n = 8) |
| 1RM (kg) | 74.8 [64.6-85.1] | 80.0 [65.8-94.2] | 46.3 [25.8-66.7] |
| 1RM (kg·mb$^{-0,67}$) | 4.55 [3.86-5.24] | 4.84 [3.72-5.97] | 2.51 [1.57-3.45] |

Data are presented as arithmetic mean and 95% confidence interval. VO2max: maximal oxygen uptake, O2pulse: oxygen uptake per heartbeat, HR: heart rate, R: respiratory exchange ratio, BMI: body mass index, 1RM: 1 repetition maximum concentrically.

**Table 6. Multiple linear regression analysis for predicting $VO_{2max}$ (mL·kg$^{-1}$·min$^{-1}$).**

| | R | R$^2$ | AdjustedR$^2$ | ΔAdjustedR$^2$ | P | SEE |
|---|---|---|---|---|---|---|
| **MEN** | | | | | | |
| **Age** | 0.58 | 0.33 | 0.33 | – | <0.001 | 5.79 |
| **Age, BMI** | 0.77 | 0.60 | 0.59 | 0.26 | <0.001 | 4.51 |
| **Age, BMI, Smoking** | 0.77 | 0.60 | 0.59 | 0.00 | <0.001 | 4.55 |
| **WOMEN** | | | | | | |
| **Age** | 0.49 | 0.24 | 0.20 | – | <0.001 | 6.27 |
| **Age, BMI** | 0.78 | 0.60 | 0.58 | 0.38 | <0.001 | 4.59 |
| **Age, BMI, Smoking** | 0.78 | 0.61 | 0.59 | 0.01 | <0.001 | 4.56 |

R: multiple regression coefficient; R$^2$: coefficient of determination; P: level of significance; SEE: standard error of estimate.

in these important prerequisites results in a consistent 6–15% lower $VO_{2max}$/$VO_{2peak}$ measurement when testing on bicycle ergometer compared to treadmill with an incline [45,77,78], a probable consequence of a lower cardiac output using cycle ergometer [64,65]. Furthermore, $VO_2$ respiratory kinetics measured with breath-by-breath systems present with significantly higher relative error and is less stable than systems using a mixing chamber [79], as well as yielding substantially lower $VO_{2max}$ measurements [80]. In the literature, there are two Norwegian reference materials [48,63] presenting aerobic capacity variables that consist of apparently healthy populations. We believe comparing our findings to the HUNT3 [48] reference material, rather than the reference material from Edvardsen and colleagues [63] opens for a more correct interpretation of differences of aerobic capacity data for several reasons. First, our study participants were recruited from the same geographical region and population base, in the middle part of Norway, as the HUNT3 participants. Second, the HUNT3 sample size is significantly larger, n = 3816 [48] vs. n = 759 [63]. Third, the latter study collected data in nine test venues throughout Norway, thus, having a relatively small population sample from the middle part of Norway when stratified for sex, age group and venue.

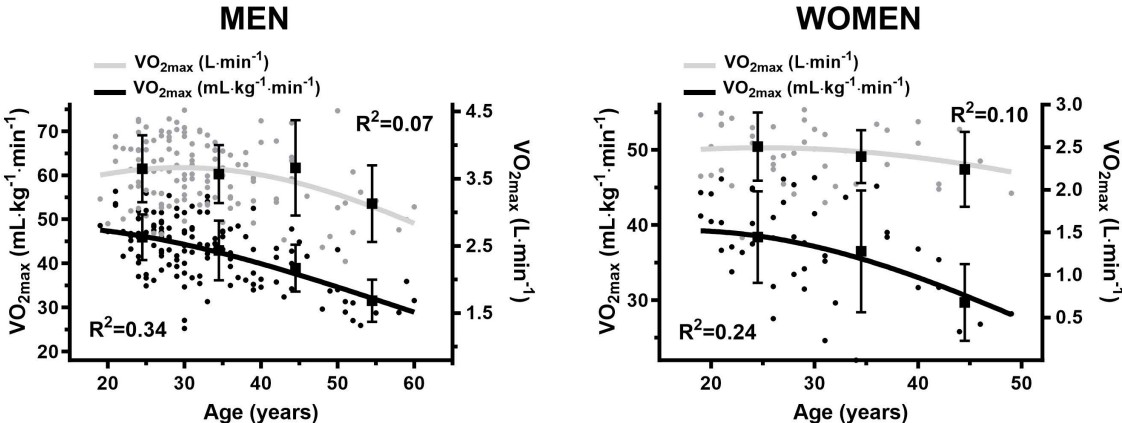

**Fig 1.** A. Decline in oxygen uptake relative to age in men. B. Decline in oxygen uptake relative to age in women.

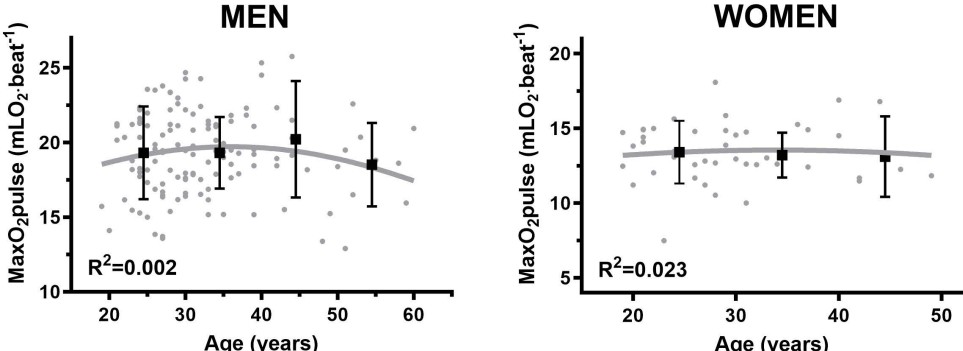

**Fig 2.** A. Changes in maximal $O_{2pulse}$ relative to age in men. B. Changes in maximal $O_{2pulse}$ relative to age in women.

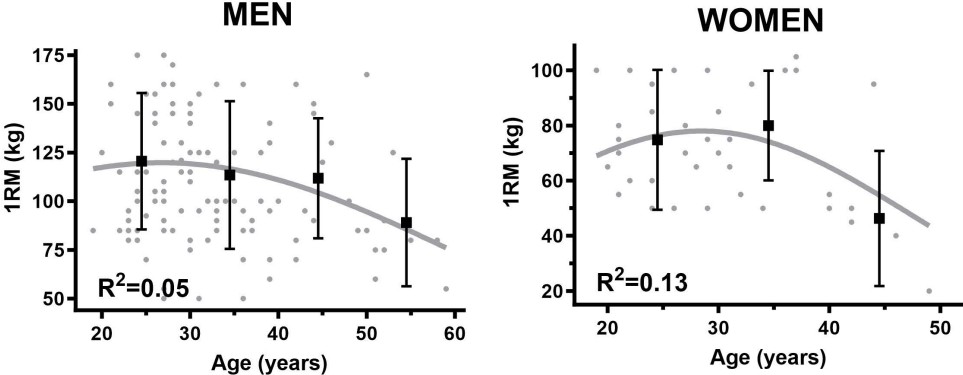

**Fig 3.** A. Decline in muscular strength in hack squat relative to age in men. B. Decline in muscular strength in hack squat relative to age in women.

## Age group differences in VO$_{2max}$

The highest relative VO$_{2max}$ in this study was found in the youngest age group (20–29 years) in both men and women, which is consistent with the HUNT3 reference material with healthy adults [48]. We observed a drop of approximately 1-Metabolic equivalent (1-MET = 3.5 mL·kg$^{-1}$·min$^{-1}$ of oxygen) per decade between the first three male age groups, which increased to a 2-MET drop between the two oldest age groups. In women, no significant change in relative VO$_{2max}$ was observed between the two youngest age groups. However, a non-significant, but clinically relevant approximately 2-MET (∼ 19%, p = 0.09) lower VO$_{2max}$ was observed between the two oldest age groups (30–39 years and 40–49 years). In comparison, the HUNT3 reference material [48] demonstrated a more linear drop of ∼ 1-MET average between and throughout all age groups among both men and women. This steeper drop in relative VO$_{2max}$ among the older SUD patients may partially be attributed to an increase in bodyweight. This could be caused by a general tendency of SUD patients leading a sedentary lifestyle [39], with a likely further decrease in physical activity with increasing age compared to a normal reference population, which will deteriorate VO$_{2max}$ even further. Adding to this challenge is the occasional use of medication causing weight gain as a side effect. Conversely, the HUNT3 reference material [48] displays increased body weight only between the two youngest age groups, after which it remains stable, with a decrease between the two most senior age groups.

Relative VO$_{2max}$ in this study was generally lower throughout the age groups, among both men and women, with an evident growing difference with increasing age compared to the HUNT3 reference material [48]. The highest differences were found in men, with an average of 2-MET lower relative VO$_{2max}$ between our findings and the HUNT3 study [48] in the first 3 age groups (20–49 years), accelerating to a more than 3-MET difference between the two oldest age groups (40–59 years). Among women, the relative VO$_{2max}$ was on average 1 MET lower compared to HUNT3 [48] in the two youngest age groups (20–39 years), increasing to a ∼ 2.5-MET difference in the oldest age group (40–49 years). However, for women one should bear in mind the relatively small sample size when interpreting the results. It is still interesting to point out tendencies in the data. The considerably lower relative VO$_{2max}$ in the two youngest male age groups (20–39 years), compared to the HUNT3 reference material [48], can likely be ascribed a sedentary lifestyle [39], heralding a low level of physical activity [43], not taxing the aerobic energy system sufficiently to even maintain their VO$_{2max}$ level. In the two oldest male age groups (40–59 years) the even higher difference in relative VO$_{2max}$ can additionally be explained by a substantially higher body weight among our participants compared to the HUNT3 material. The pattern for women is the same, although smaller differences was demonstrated.

Compared to HUNT3 [48], male SUD patients in the first 3 age groups (20–49 years) had a relative VO$_{2max}$ corresponding to being in excess of 20 years older than their actual age, further attenuated to a more than 30 years difference in the oldest age group (50–59 years). A comparable trend is apparent among women, where the two youngest age groups (20–39 years) showed a relative VO$_{2max}$ corresponding to more than 10 years of ageing compared to HUNT3 [48], which further deteriorated to a 25 year difference in the oldest age group (40–49 years).

No significant differences in absolute VO$_{2max}$ (L·min$^{-1}$) were found between any of the age groups among men and women in the SUD data. This is probably due to the limited sample size, since there is an evident tendency towards lower values among the oldest age groups in both sexes. The decline rate between the two oldest age groups of men- (40–49 yrs. and 50–59 yrs.) and women (30–39 yrs. and 40–49 yrs.) were approximately 12% and 6%, respectively. This is comparable to the tendency in the HUNT3 reference material [48] in similar age groups. However, absolute VO$_{2max}$ was in our material substantially lower in each age group compared to the HUNT3 material [48], being on average ∼ 14% lower among both men and women. In other words, the youngest SUD patients had an absolute VO$_{2max}$ similar to HUNT3 participants 20- and 30 years their seniors, in women and men, respectively.

Scaled VO$_{2max}$ deteriorated by a similar trajectory as relative VO$_{2max}$, in both sexes in the SUD population and is comparable to the HUNT3 reference material [48]. However, the SUD measurements were approximately 10 and 15% lower in each age group for women and men, respectively, and increased to in excess of 20% in the oldest age groups. This

difference is lower than in the relative $VO_{2max}$ ($\sim$ 25%), and ascribed the effect of bodyweight being less dominant in scaled data.

### Age groups differences in maximal $O_{2pulse}$

There were no significant differences in maximal $O_{2pulse}$ between subsequent age groups among men and women in our data. However, our slope trajectory demonstrated a $\sim$ 7% non-significant (probably attributed limited sample size) reduction in $O_{2pulse}$ between the two oldest male age groups (40–49 yrs. and 50–59 yrs.). This drop in maximal $O_{2pulse}$ corresponds well with the findings in the HUNT3 reference material [48] comparing similar age groups. We observed an average $\sim$ 9% and $\sim$ 12% lower maximal $O_{2pulse}$ in each SUD patient age group compared to the HUNT3 material [48] in women and men, respectively.

### Age group differences in 1RM

In our data, there were no significant differences in 1RM between the youngest age groups in both sexes. Between the oldest male age groups (40–49 yrs. and 50–59 yrs.), a clinically relevant $\sim$ 20% (non-significant) reduction in 1RM and $\sim$ 25% in scaled 1RM were observed, with corresponding reductions of $\sim$ 40 and $\sim$ 50% between our oldest age groups among women (30–39 yrs. and 40–49 yrs.). Direct comparison between our findings and other research is difficult given the vast diversity in methodology in the literature, e.g., muscular strength measured as a ratio: weight pushed in leg press/ body weight, ordinary leg press (kg), peak torque (Nm) in knee extension or leg extension denoted as power (W) [53,81–84]. Additionally, leg press and single joint movements hold several disadvantages compared to hack squat. Hack squat initiates from an upright position, with a movement cycle more comparable to walking, thus, enabling a more functional movement pattern. Furthermore, it is more demanding on the stabilization of the truncus, which is also more in resemblance with walking [85]. Regardless of measuring method, the general tendency in previous studies is either a lowering of muscular strength between each consecutive older age group, or peaking in muscular strength in the 20–39 yrs. age groups, followed by an 8–15% decline between subsequent age groups [53,81–83,86,87]. Comparing our findings in muscular strength, within each age group, to a normative material from the U.S. [53] demonstrated that our data was on average $\sim$ 15% and 25% lower in the younger age groups in women and men, respectively, diverging to $\sim$ 48% and $\sim$40% in the oldest age groups when applying identical measuring methodology (ratio: 1RM/ bodyweight). However, presenting data this way is highly impacted by bodyweight, enabling a biased interpretation given the increasing of body weight throughout all age groups in our data, contradictory to the bodyweight trajectory in the HUNT3 reference material [48]. Nevertheless, results in a previous study from this research group [47], measuring 1RM in hack squat, demonstrated a 30% and 33% lower 1RM in men and women (all ages combined), respectively, compared to healthy controls.

### Sex differences in $VO_{2max}$, maximal $O_{2pulse}$ and 1RM

For a more accurate comparison of the weighted average between men and women only the 3 youngest age groups are used in the analyzes (20−29 years, 30−39 years, 40−49 years), since no women older than 49 years were included. Women make up only one third of patients in SUD treatment, which is consistent with previous research [88]. In this study women presented with $\sim$ 17% and $\sim$ 33% lower relative ($mL \cdot kg^{-1} \cdot min^{-1}$) – and absolute ($L \cdot min^{-1}$) $VO2_{max}$ than men, respectively, and $\sim$ 21% lower scaled ($mL \cdot kg^{-0.67} \cdot min^{-1}$) $VO_{2max}$ accounting for differences in bodyweight enabling a more accurate comparison [60,89]. This is expected, and explained by a lower hemoglobin count, smaller heart and lungs in women [90]. In addition, tentative data in a parallel ongoing study in our clinic display a lower self-reported physical activity level in women than in men, which could in part explain the lower cardiorespiratory fitness. However, one should consider the inherent possibility of misreporting when utilizing self-reporting [91,92]. HUNT3 reference data [48] display sex differences in $VO_{2max}$ coinciding fairly well with the findings in our SUD patient population, i.e., women in HUNT3 show

a ~ 19%, ~ 34% and ~ 24% lower relative- absolute- and scaled $VO_{2max}$, respectively. This is also in agreement with a previous study (n = 44) on SUD patients from our clinic [47].

Sex differences in our maximal $O_{2pulse}$ data corresponds well with those of healthy Norwegians [48], i.e., women scoring approximately 32% lower than men, which can be explained by women generally having a smaller heart, therefore a smaller left ventricular stroke volume, as well as a lower blood hemoglobin concentration [93]. Women also have a smaller area of peripheral oxygen extraction (i.e., mitochondrial surface area), evident when utilizing dimensional scaling, connected to women generally being of shorter stature than men [78].

In muscular strength women obtained a roughly 38% lower 1RM in hack squat than men, which is in agreement with previous findings among both SUD patients as well as healthy controls in our clinic using hack squat [47]. A U.S. normative material [53], including apparently healthy persons, showed ~ 25% lower muscular strength among women aged 20–49 years compared to men, reported as the ratio between 1RM in leg press/ body weight, which is comparable to our findings (~ 24% lower) when applying similar calculations. Another U.S. normative material [83], with an apparently healthy population, demonstrated a ~ 33% lower peak torque (Nm) among women (20–49 years) performing a concentric knee extension (i.e., using m. quadriceps). Reviews [93,94] point out an expected 30–50% lower muscular strength among women in the lower extremities compared to men. Sex differences in muscular strength are caused by differences in muscle biology (i.e., men having a larger muscle- cross sectional area and fiber density, higher levels of testosterone) [93,95] and biomechanical traits (i.e., men having longer lever arms yielding higher torque) [78].

### Risk of disease and all-cause mortality related to low- $VO_{2max}$, maximal $O_{2pulse}$ and 1RM

Three percent of our male SUD participants displayed $VO_{2max}$ values below 8-MET (28 mL·kg$^{-1}$·min$^{-1}$) which is associated with higher prevalence of all-cause mortality and cardiovascular disease in healthy populations [24]. It is noteworthy that more than 80% of these male SUD participants were in the two youngest age groups (20−39 yrs.), and that low MET was present already in the 30−39 yrs. group for half of the cases contrary to HUNT3 data, where approximately one percent displayed a MET associated with elevated risk [48]. More importantly, in the HUNT3 material none were under 50 years, with the majority (80%) being in the most senior age groups (60−90 yrs.). Only one woman in our study had an aerobic capacity below 6-MET (21 mL·kg$^{-1}$·min$^{-1}$), the value associated with increased risk of cardiovascular disease and all-cause mortality in women [24]. The prevalence of MET values associated with increased risk of poor health and mortality is lower among women than in men, which coincide with HUNT3 findings. Furthermore, a 1-MET (3.5 mL·kg$^{-1}$·min$^{-1}$) higher relative $VO_{2max}$ is associated with a 12% and 17% improved survival rate in men and women, respectively [26,96]. This is supported by recent findings where every 1-MET higher $VO_{2max}$ was associated with 11% lower relative risk for all-cause mortality [97]. When extrapolating the current decline rate in the SUD population, the average $VO_{2max}$ for men > 60 years appears to be below the cut-off score, signaling increased risk of ill health and all-cause mortality [24]. The tendency was similar for women.

Peak $O_{2pulse}$ has also been demonstrated as a forecaster of cardiovascular disease and mortality [29,98], where a low peak $O_{2pulse}$ signifies a worse cardiorespiratory function, and by the same token better when it is higher [99]. Approximately three percent of our male participants presented with a maximal $O_{2pulse}$ < 13.5 mL·beat$^{-1}$, which in a Finnish study with males signified a 2.45-fold higher relative risk (RR) of death caused by cardiac heart disease, and a 1.79-fold higher RR in overall death, compared to those with a maximal $O_{2pulse}$ > 17.8 mL·beat$^{-1}$ [29]. Furthermore, $O_{2pulse}$ at maximal exercise in a male population was shown to have a linear inverse relationship with all-cause mortality [98]. When categorizing our male participants by the maximal $O_{2pulse}$ quartiles as displayed in a paper by Laukkanen and colleagues [98] ~ 20% are in the highest maximal $O_{2pulse}$ (21.9–42.7 mL·beat$^{-1}$) quartile, ~ 60% are evenly distributed in the two following quartiles (19.3–21.9 mL·beat$^{-1}$ and 16.8–19.3 mL·beat$^{-1}$) and ~ 20% in the lowest quartile (6.4–16.8 mL·beat$^{-1}$). Attenuated maximal $O_{2pulse}$ across quartiles denotes an increased hazard ratio compared to the previous quartile.

While a plethora of research has consistently cemented the strong association between cardiorespiratory variables (e.g., $VO_{2max}$ and maximal $O_{2pulse}$) and adverse health outcomes and mortality over the past 4–5 decades, more contemporary findings demonstrate that maximal muscular strength holds many of the same prognostic properties. Similar to cardiorespiratory variables (e.g., VO2max and maximal O2pulse), maximal muscular strength is inversely connected to several lifestyle related diseases, such as type II diabetes [32], cancer [33], cardiovascular disease [30,31] as well as being an independent forecaster of all-cause mortality [31,35–37]. Thus, the lower muscular strength capacity observed in the present study likely constitutes yet a health-related risk factor for these patients. The generally lower- relative $VO_{2max}$, maximal $O_{2pulse}$ and 1RM throughout all age groups, compared to reference materials of healthy age-matched populations [48,53] are probably important factors in explaining the higher level of morbidity and comorbidity, as well as the15–30 years shorter life expectancy in the SUD population [39,41,42].

Selection bias is an inherent risk in all data collections involving physical testing to volitional exertion, where those least fit could chose not to partake. This potential bias is probably amplified among SUD patients where the fraction of less fit individuals is considerably higher than in a healthy population. Additionally, the least fit SUD patients are often unable to complete vigorous testing due to physical- or mental constraints and side effects of medication. Consequentially, the actual $VO_{2max}$, maximal $O_{2pulse}$ and 1RM in SUD patients is probably lower than presented in our data, potentially making the difference between our results and other reference materials with healthy participants even bigger. Accordingly, the risk of lifestyle related diseases and all-cause mortality is probably underestimated in our data.

### Clinical implications

Physical activity and exercise training has generally been implemented as an adjunctive intervention in SUD clinics; however, adherence remains low and attrition rates are high. Moreover, the physical activity provided in clinical settings appears to lack structure and systematic progression and is insufficiently demanding to adequately challenge the aerobic and neuromuscular systems, thereby limiting their potential to produce clinically meaningful improvements in physical fitness [57]. Given the increased risk of lifestyle-related diseases associated with poor $VO_{2max}$ and muscular strength, our findings underscore the importance of structured physical training being an integrated activity in rehabilitation and health promotion programs in SUD populations.

### Strengths and limitations

A key strength of this study is the use of direct measurement of $VO_{2max}$ and maximal $O_{2pulse}$ by the use of treadmill testing, employing the Vyntus CPX system with a mixing chamber. This approach offers a more accurate assessment of aerobic capacity compared to prediction models or submaximal testing methods. All included participants are diagnosed with SUD applying the (ICD-10, F10-F19). However, the study is limited by a relatively small sample size, particularly in the oldest age groups among both men and women. This limitation hinders the ability to perform robust subgroup analyses and reduces the generalizability of our findings. Additionally, the possibility of a selection bias must be acknowledged, as physical testing to volitional exhaustion may deter participation among individuals with lower fitness levels. To enhance the validity and applicability of these findings, future studies should involve larger and more diverse samples drawn from various populations with SUD.

### Conclusions

This study represents the most comprehensive reference material on key aerobic- and muscular strength variables in patients with a SUD and constitutes a significant and more complete addition to the existing literature. Patients with SUD generally demonstrated lower aerobic capacity across all age groups compared to healthy age-matched Norwegians, particularly among men, translating to a three-fold higher prevalence of low aerobic capacity. Also, maximal muscular strength was lower compared to healthy controls. Notably, these deficits were observed at a considerable younger age than in healthy controls.

## Supporting information

**S1 Raw Data. Raw data.**

(SAV)

## Acknowledgments

The authors thank the employees in our training facility for their excellent contribution in conducting parts of the testing in the study.

## Author contributions

**Conceptualization:** Henrik Loe, Grete Flemmen, Ulrik Wisløff, Mats Peder Mosti.

**Data curation:** Henrik Loe, Grete Flemmen.

**Formal analysis:** Henrik Loe.

**Funding acquisition:** Henrik Loe, Grete Flemmen.

**Investigation:** Henrik Loe, Grete Flemmen.

**Methodology:** Henrik Loe, Mats Peder Mosti.

**Project administration:** Henrik Loe, Grete Flemmen.

**Resources:** Henrik Loe.

**Supervision:** Henrik Loe.

**Visualization:** Henrik Loe.

**Writing – original draft:** Henrik Loe.

**Writing – review & editing:** Grete Flemmen, Ulrik Wisløff, Helle Wessel Andersson, Mats Peder Mosti.

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
