## [Decision Letter · Decision Letter 0]

14 Oct 2025

PONE-D-25-47064Aerobic capacity and muscular strength in 179 Norwegian men and women aged 20-59 years with a substance use disorder: A cross-sectional studyPLOS ONE

Dear Dr. Loe,

Thank you for submitting your manuscript to PLOS ONE. After careful consideration, we feel that it has merit but does not fully meet PLOS ONE’s publication criteria as it currently stands. Therefore, we invite you to submit a revised version of the manuscript that addresses the points raised during the review process.

We look forward to receiving your revised manuscript.

Kind regards,

Mário Espada, PhD

Academic Editor

PLOS ONE

Journal Requirements:

“This work is supported by Ekstrastiftelsen Helse og Rehabilitering (Stiftelsen DAM) grant number 2019/F0235306 and The Joint Research Committee between St.Olavs hospital and the Faculty of Medicine and Health Sciences, Norwegian University of Science and Technology   grant number 2023/36515”

Reviewers' comments:

Reviewer's Responses to Questions

**Comments to the Author**

1. Is the manuscript technically sound, and do the data support the conclusions?

Reviewer #1: Yes

Reviewer #2: Yes

Reviewer #3: Yes

2. Has the statistical analysis been performed appropriately and rigorously?

Reviewer #1: Yes

Reviewer #2: Yes

Reviewer #3: Yes

3. Have the authors made all data underlying the findings in their manuscript fully available?

Reviewer #1: No

Reviewer #2: No

Reviewer #3: Yes

4. Is the manuscript presented in an intelligible fashion and written in standard English?

Reviewer #1: Yes

Reviewer #2: Yes

Reviewer #3: Yes

5. Review Comments to the Author

Reviewer #1: The research question is clearly defined, and the study design—a cross-sectional assessment using direct cardiopulmonary exercise testing (CPET) and muscular strength testing—is suitable to meet the stated objectives. The methodology is described in detail, with careful calibration of equipment and the use of standardized testing protocols. The statistical analysis was correctly selected and applied (ANOVA, Bonferroni correction, t-tests, Pearson correlations), ensuring the reliability of the results. The findings show consistently lower aerobic capacity and muscular strength in SUD patients compared to healthy reference populations. The conclusions are well aligned with the data presented. Data Availability While the manuscript states that “all relevant data are within the manuscript and its supporting information files” and that data will be made available upon acceptance, this does not fully comply with the PLOS ONE Data Availability Policy. The raw data underlying the results (e.g., individual-level data points behind the means, standard deviations, and figures) are not currently included in the submission. To ensure transparency and reproducibility, the authors should provide anonymized individual-level data in a public repository or as supplementary files. Language and Presentation The manuscript is clearly written in standard English. The language is professional, understandable, and suitable for publication. Minor typographical and formatting errors can be corrected during revision, but overall the text is clear, concise, and unambiguous. Additional Comments The study’s strengths include its relatively large sample of patients with SUD, the use of objective and direct measurements of VO₂max and muscular strength, and the comparison with large normative reference datasets. Limitations are acknowledged, including small subgroup sizes in older age groups, potential selection bias (where less fit participants may have opted out), and the lack of female participants above 49 years. The authors may consider further discussion on the clinical implications of their findings for rehabilitation and health promotion programs in SUD populations. Overall Recommendation This is a well-conducted study that provides valuable new reference data. The main revision needed is compliance with the PLOS ONE Data Availability Policy by providing the raw data underlying the findings. Once this is addressed, the manuscript will be suitable for publication.

Reviewer #2: Manuscript ID: PONE-D-25-47064

Title: Aerobic capacity and muscular strength in 179 Norwegian men and women aged 20–59 years with a substance use disorder: A cross-sectional study

This study assesses aerobic capacity (VO₂max) and muscular strength (1RM) in adults with substance use disorders (SUD) using direct physiological testing. A total of 179 inpatients (aged 20–59 years) from a Norwegian clinic were compared with normative data from healthy populations. The results show substantially lower aerobic capacity and strength across all age groups, highlighting a potential link between physical deconditioning and elevated health risks in this population.

The topic is relevant and timely. Direct measurement of VO₂max and 1RM in people with SUD is uncommon, and this dataset provides valuable clinical and public-health insight.

1. Strengths

- The study addresses a clear evidence gap and offers meaningful clinical reference data for a vulnerable group.

- The use of direct CPET with calibration and a standardized strength protocol adds credibility. The application of allometric scaling is also commendable.

- Ethical approval, recruitment period, and trial registration are clearly reported.

- Tables and descriptions are generally well structured, and the discussion connects the results with existing literature.

2. Major Comments

1. Selection bias and representativeness

Because participation required maximal testing, the sample may represent the fitter segment of the SUD inpatient population. Please provide the number of screened, excluded, or non-participating individuals and, if possible, basic characteristics (e.g., age, sex, BMI) to help readers assess representativeness.

2. Statistical adjustment for confounders

The analyses are largely descriptive. Consider including regression models that adjust for potential confounders (age, BMI, smoking, type/duration of substance use, medications, comorbidities). Even simple adjusted models would enhance analytical rigor.

3. Limited subgroup power

Some subgroups (notably older women) are small. Report confidence intervals and effect sizes to aid interpretation and explicitly discuss this limitation.

4. Comparability of reference values

The use of the HUNT3 dataset as reference data is appropriate geographically, but readers need clarity on differences in testing protocols, calibration, and population characteristics. A brief summary table or paragraph would help contextualize the comparisons.

5. Data availability statement

PLOS ONE requires data to be available at publication unless legal or ethical barriers apply. Please clarify exactly how and when data will be shared (e.g., through an anonymized repository, with DOI or controlled access).

6. Adverse events and test completion

Given the maximal CPET and 1RM protocols, please report whether any participants failed to complete testing or experienced adverse events, and how these cases were handled.

7. Instrumentation note

The manuscript mentions a 5.5% error at very high ventilation rates (VE > 180 L/min). Indicate how many participants this affected and whether this could meaningfully influence results.

3. Minor and Editorial Comments

- Report 95% confidence intervals and effect sizes (Cohen’s d, η², etc.) for key comparisons.

- Verify assumptions of normality and variance homogeneity; note if any nonparametric approaches were used when appropriate.

- Streamline sections of the discussion to reduce redundancy and focus on the main messages.

- Adding scatterplots (e.g., VO₂max vs. age, strength vs. age) would help visualize inter-individual variability.

- Clarify technical terms (e.g., “O₂ pulse,” “allometric scaling”) for non-specialist readers.

- A light proofreading for grammar and flow would further improve readability.

4. Are the Conclusions Supported by the Data?

In general, yes. The observed reductions in VO₂max and strength among participants with SUD are consistent with prior knowledge. However, the conclusions should remain descriptive rather than causal, given the cross-sectional design and absence of adjustment for potential confounders. Emphasizing this limitation would make the interpretation more balanced and scientifically precise.

5. Recommendation : Major Revision

This is a valuable and well-conducted study that could make a meaningful contribution once some clarifications and analytical refinements are addressed.

The issues identified—mainly regarding statistical adjustment, transparency about sample selection, and data-sharing details—are entirely fixable within a reasonable revision process.

With these improvements, the manuscript would be strong enough for publication in PLOS ONE.

Reviewer #3: I would like to appreciate the efforts of the authors in implementing the project and writing this article “Aerobic capacity and muscular strength in 179 Norwegian men and women aged 20-59 years with a substance use disorder: A cross-sectional study”.

The main objective of this study was to contribute to the establishment of reference values for directly measured parameters of aerobic capacity and muscular strength in patients with SUD and to compare these data with normative values consisting of apparently healthy age-appropriate controls.

The submitted article contains valuable data on aerobic capacity and muscular strength in patients with addiction and provides the most extensive set of reference values in this population.

I have the following comments or questions:

- How was it determined in the selection of participants who is able to perform exhaustive endurance and strength training and total testing. I recommend describing the selection criteria in more detail.

- I believe that the study would benefit greatly from reporting the differences in measured VO2max values in the general population This would facilitate readers interpretation of the results given

- I recommend to deepen the discussion on the possible biochemical and psychosocial causes of the observed deficits.

6. PLOS authors have the option to publish the peer review history of their article (what does this mean?). If published, this will include your full peer review and any attached files.

**Do you want your identity to be public for this peer review?** For information about this choice, including consent withdrawal, please see our Privacy Policy.

Reviewer #1: **Yes:** Hala Awad Ahmed

Reviewer #2: No

Reviewer #3: No

---

## [Author Response · Author response to Decision Letter 1]

25 Mar 2026

Dear Editor,

Thank you for your patience with our delay, and for giving us the opportunity to revise and re-submit an improved version of our manuscript. All comments and concerns from the editor and reviewers have been addressed with corresponding amendments in the manuscript. We thank you and the reviewers for valuable comments, surely making the manuscript better. We do hope that the revised version of the manuscript is satisfactory for publication in PLOS ONE.

Sincerely,

Henrik Loe

Response to Journal and Reviewers:

Journal:

The amended role of funder statement is now included in the cover letter as requested.

Thank you for the information on the potentially time consuming data availability process. We prefer to make the data freely accessible if the manuscript is accepted for publication.

Reviewer #1:

Thank you for taking the time to review our manuscript. We believe that your suggestions and comments have contributed to improve it. Please find the changes highlighted in the revised version of the manuscript.

• Addressing your concern regarding data availability: In line with PLOS ONE guidelines, all individual raw data underlying the results will be made freely available if the manuscript is accepted for publication. Data Availability Policy, so that all statistical analysis can be scrutinized.

• Response to your input on clinical implications. This is clearly a relevant and important factor and has now been addressed in an added section under the Discussion (lines 554-563)

Reviewer #2:

Thank you for your comprehensive feedback. Your comments and concerns are hopefully addressed adequately. Please find the changes highlighted in the revised version of the manuscript.

2. Major Comments

1. Selection bias and representativeness

Comment: Because participation required maximal testing, the sample may represent the fitter segment of the SUD inpatient population. Please provide the number of screened, excluded, or non-participating individuals and, if possible, basic characteristics (e.g., age, sex, BMI) to help readers assess representativeness.

Reply: Those who declined to partake when eligible for participation (Non-participants) and inclusion rate are now included under the heading: Settings and participants in the methods section (Lines 88-93). Unfortunately, basic characteristics were not collected from those who declined.

As you point out, participants may represent the fitter segment. This has been noted in the strength and limitation section (Lines 573-575), and in the discussion with a brief elaboration of consequences (Lines 543-550).

2. Statistical adjustment for confounders

Comment: The analyses are largely descriptive. Consider including regression models that adjust for potential confounders (age, BMI, smoking, type/duration of substance use, medications, comorbidities). Even simple adjusted models would enhance analytical rigor.

Reply: Thank you for this notice. We agree that a regression analysis for the key variable VO2max will improve analytical rigor. This has been included in a new table (Table 6) in the Results section (Lines 306-311). This analysis is also described in the Statistical analysis section (Lines 218-219). Unfortunately, we did not have access to the patients’ medical records and were therefore unable to assess individual substance use, medication, or comorbidities.

3. Limited subgroup power

Comment: Some subgroups (notably older women) are small. Report confidence intervals and effect sizes to aid interpretation and explicitly discuss this limitation.

Reply: Confidence intervals are now reported in the revised version of table 4 and 5. Overall effect sizes from the one-way ANOVA are reported at the end of the paragraphs describing “age group differences” in the results section (Lines 268-269 and 291-293).

4. Comparability of reference values

Comment: The use of the HUNT3 dataset as reference data is appropriate geographically, but readers need clarity on differences in testing protocols, calibration, and population characteristics. A brief summary table or paragraph would help contextualize the comparisons.

Reply: Thank you for this suggestion. This has been rectified in a new paragraph in the Methods section named HUNT 3 reference material (Lines 147-154). We have also provided a new table (Table 2) summarizing basic characteristics (age group, sample size, VO2max, body mass, R-value).

5. Data availability statement

Comment: PLOS ONE requires data to be available at publication unless legal or ethical barriers apply. Please clarify exactly how and when data will be shared (e.g., through an anonymized repository, with DOI or controlled access).

Reply: In response to data availability. If the manuscript is accepted for publication all individual raw data underlying the results will be made freely available to be in compliance with the PLOS ONE Data Availability Policy, so that all statistical analysis can be scrutinized.

6. Adverse events and test completion.

Comment: Given the maximal CPET and 1RM protocols, please report whether any participants failed to complete testing or experienced adverse events, and how these cases were handled.

Reply: No adverse events occurred and everyone who partook completed tests. This is now specified under the heading: Settings and participants (Lines 88-93) in the Methods.

7. Instrumentation note.

Comment: The manuscript mentions a 5.5% error at very high ventilation rates (VE > 180 L/min). Indicate how many participants this affected and whether this could meaningfully influence results.

Reply: Two participants exceeded the predefined acceptance for error in the highest minute ventilations. This is now addressed in the Methods in the CPET section (Lines 138-140).

3. Minor Editorial Comments.

Comment: Report 95% confidence intervals and effect sizes (Cohen’s d, η², etc.) for key comparisons.

Reply: Amended as suggested. As noted in our previous reply, confidence intervals are now reported in the revised version of table 4 and 5. Overall effect sizes from the one-way ANOVA are reported at the end of the paragraphs describing “age group differences” in the results section (Lines 268-269 and 291-293).

Comment: Verify assumptions of normality and variance homogeneity; note if any nonparametric approaches were used when appropriate.

Reply: The assumption of normally distributed data was supported by QQ-plots, and variance homogeneity by Levene’s test. No nonparametric approaches were applied. This information is now added in the Statistical analysis section (Lines 211-212).

Comment: Streamline sections of the discussion to reduce redundancy and focus on the main messages.

Reply: We have made an effort to streamline sections of the discussion.

Comment: Adding scatterplots (e.g., VO2max vs. age, strength vs. age) would help visualize inter-individual variability.

Reply: Thank you for this notice. All figures have been revised accordingly, now including scatterplots.

Comment: Clarify technical terms (e.g., “O2 pulse,” “allometric scaling”) for non-specialist readers

Reply: In the revised manuscript, this is addressed in the Introduction- (Lines 50-52) and Allometric scaling paragraph of the Methods (Lines 200-201).

Comment: A light proofreading for grammar and flow would further improve readability.

Reply: Proofreading has been performed.

4. Are the conclusions supported by the data?

Comment: …the conclusions should remain descriptive rather than causal, given the cross-sectional design and absence of adjustment for potential confounders.

Reply: Revised as suggested. The conclusion is made more descriptive in the revised manuscript.

Reviewer #3:

Thank you for reviewing our manuscript. Your comments and concerns are hopefully addressed adequately. Please find the changes highlighted in the revised version of the manuscript.

Comment: How was it determined in the selection of participants who is able to perform exhaustive endurance and strength training and total testing. I recommend describing the selection criteria in more detail.

Reply: This is now elaborated on under the heading: Settings and participants in the Methods (Lines 88-93). In brief, everyone who fulfilled the inclusion criteria and were able to complete the physical tests were regarded as eligible.

Comment: I believe that the study would benefit greatly from reporting the differences in measured VO2max values in the general population. This would facilitate readers interpretation of the results given.

Reply: We do agree that key descriptives from the HUNT 3 study will make comparisons more accessible. This has been rectified in the Methods section, where a new paragraph and table (Table 2) display test specifics and characteristics of HUNT 3, respectively (Lines 147-154).

Comment: I recommend to deepen the discussion on the possible biochemical and psychosocial causes of the observed deficits.

Reply: These are no doubt important and interesting factors to discuss. Unfortunately, we have not collected any biochemical or psychosocial data, but we have addressed the possible causes of the observed deficits in the third paragraph of the discussion (Lines 375-381). This is certainly a relevant issue and should be implemented in future research.

---

## [Decision Letter · Decision Letter 1]

2 Apr 2026

PONE-D-25-47064R1Aerobic capacity and muscular strength in 179 Norwegian men and women aged 20-59 years with a substance use disorder: A cross-sectional studyPLOS One

Dear Dr. Loe,

Thank you for submitting your manuscript to PLOS ONE. After careful consideration, we feel that it has merit but does not fully meet PLOS ONE’s publication criteria as it currently stands. Therefore, we invite you to submit a revised version of the manuscript that addresses the points raised during the review process.

We look forward to receiving your revised manuscript.

Kind regards,

Mário Espada, PhD

Academic Editor

PLOS One

Journal Requirements:

Additional Editor Comments:

Dear Authors,

Congratulations on your work.

Please revise the manuscript in light of the reviewers' minor suggestions.

Thank you.

Best regards.

Reviewers' comments:

Reviewer's Responses to Questions

**Comments to the Author**

1. If the authors have adequately addressed your comments raised in a previous round of review and you feel that this manuscript is now acceptable for publication, you may indicate that here to bypass the “Comments to the Author” section, enter your conflict of interest statement in the “Confidential to Editor” section, and submit your "Accept" recommendation.

Reviewer #2: (No Response)

Reviewer #3: All comments have been addressed

2. Is the manuscript technically sound, and do the data support the conclusions?

Reviewer #2: Yes

Reviewer #3: Yes

3. Has the statistical analysis been performed appropriately and rigorously? 

Reviewer #2: Yes

Reviewer #3: Yes

4. Have the authors made all data underlying the findings in their manuscript fully available?

Reviewer #2: No

Reviewer #3: Yes

5. Is the manuscript presented in an intelligible fashion and written in standard English?

Reviewer #2: No

Reviewer #3: Yes

6. Review Comments to the Author

Reviewer #2: The revised manuscript substantially addresses the concerns from the first review round and is now suitable for PLOS ONE with minor revisions.

- Strengths: Added non-participant count (87% inclusion), HUNT3 reference section/Table 2, CIs in Tables 4-5, effect sizes (η²), scatterplots, and VO2max regression (Table 6). Conclusions appropriately descriptive.

- Remaining issues:

*Data availability: Specify repository (e.g., Figshare/Zenodo), DOI, or exact mechanism.

*Tables 4/5: Reformatting needed—align columns, fit on one page if possible, consistent CI notation (e.g., 45.9 [44.5–47.3]).

- Minor language/formatting (e.g., "was tested" → "were tested"; effect sizes like "20.329 0,327" need decimal fix).

**Recommendation**: Minor revision. Core science solid and valuable.

Reviewer #3: The work is very interesting, all my comments have been incorporated and resolved. I have no further questions or comments.

7. PLOS authors have the option to publish the peer review history of their article (what does this mean?). If published, this will include your full peer review and any attached files.

Reviewer #2: No

Reviewer #3: No

---

## [Author Response · Author response to Decision Letter 2]

13 Apr 2026

Dear Editor,

Thank you for giving us the opportunity to re-submit a minor revision of our manuscript. Comments and concerns from the reviewers have been addressed with corresponding amendments in the manuscript. We do hope that the revised version of the manuscript is satisfactory for publication in PLOS ONE.

Sincerely,

Henrik Loe

Response to Journal and Reviewers:

Reviewer #2:

Comment: Data availability: The commitment to share raw data "freely if accepted" aligns with PLOS ONE policy in principle, but lacks specifics on repository (e.g., Figshare, Zenodo), DOI, or timeline—please clarify this before acceptance to fully comply

Reply: All relevant data will be included in Supporting Information files if manuscript is accepted for publication, an option in compliance with PLOS ONE data availability policy.

Minor edits:

Comments: Table 4 (and likely Table 5 for women) needs reformatting for clarity—ensure columns align properly, full data visible on one page if possible, and consistent CI notation (e.g., 45.9 [44.5–47.3])

Reply: These concerns have now been rectified in Table 4 (lines 270-273) and Table 5 (lines 293-296).

Comment: Some phrasing remains awkward (e.g., "Patients that fulfilled the inclusion criteria, and agreed to participate, was tested" should be "were tested"; "oxygen pumpedheartbeat" needs spacing/formatting).

Reply: The verb (was) is corrected to plural (were) (line 89). “oxygen pumped/heartbeat” changed to “oxygen pumped per heartbeat” (line 52)

Comment: Ensure consistent formatting for CIs and effect sizes across text/tables (e.g., η² values like "20.329 0,327" appear malformed).

Reply: Consistent formatting for CI implemented in Table 4 (lines 270-273) and Table 5 (lines 293-296) and effect sizes (lines 268-269 and lines 291-292).

Comment: The CPET ventilation note (two participants >180 L/min) is clear but could be more concise.

Reply: Amended the sentence with a more concise formulation as suggested (lines 138-139).

Journal:

No references have been added or retracted.

---

## [Decision Letter · Decision Letter 2]

21 Apr 2026

Aerobic capacity and muscular strength in 179 Norwegian men and women aged 20-59 years with a substance use disorder: A cross-sectional study

PONE-D-25-47064R2

Dear Dr. Henrik Loe,

We’re pleased to inform you that your manuscript has been judged scientifically suitable for publication and will be formally accepted for publication once it meets all outstanding technical requirements.

Kind regards,

Mário Espada, PhD

Academic Editor

PLOS One

---

## [Editor Report · Acceptance letter]

PONE-D-25-47064R2

PLOS One

Dear Dr. Loe,

I'm pleased to inform you that your manuscript has been deemed suitable for publication in PLOS One. Congratulations! Your manuscript is now being handed over to our production team.

Kind regards,

on behalf of

Dr. Mário Espada

Academic Editor

PLOS One